# Effects of Age and MPTP-Induced Parkinson’s Disease on the Expression of Genes Associated with the Regulation of the Sleep–Wake Cycle in Mice

**DOI:** 10.3390/ijms25147721

**Published:** 2024-07-14

**Authors:** Ekaterina I. Semenova, Margarita M. Rudenok, Ivan N. Rybolovlev, Marina V. Shulskaya, Maria V. Lukashevich, Suzanna A. Partevian, Alexander I. Budko, Maxim S. Nesterov, Denis A. Abaimov, Petr A. Slominsky, Maria I. Shadrina, Anelya Kh. Alieva

**Affiliations:** 1National Research Centre “Kurchatov Institute”, 2 Kurchatova Sq., 123182 Moscow, Russia; rudenokmm.img@yandex.ru (M.M.R.); ivan.rybolovlev@yandex.ru (I.N.R.); shulskaya.m@yandex.ru (M.V.S.); farouki@mail.ru (M.V.L.); s.partev@yandex.ru (S.A.P.); budko_ai@rrcki.ru (A.I.B.); paslominsky@bk.ru (P.A.S.); maria.i.shadrina@yandex.ru (M.I.S.); anelja.a@gmail.com (A.K.A.); 2Scientific Center for Biomedical Technologies of the Federal Biomedical Agency of Russia, 119435 Krasnogorsk, Russia; mdulya@gmail.com; 3Research Center of Neurology, Volokolamskoye Shosse 80, 125367 Moscow, Russia; abaidenis@yandex.ru

**Keywords:** Parkinson’s disease, MPTP model, neurodegeneration, sleep–wake cycle, histamine, circadian genes, gene expression, transcriptome

## Abstract

Parkinson’s disease (PD) is characterized by a long prodromal period, during which patients often have sleep disturbances. The histaminergic system and circadian rhythms play an important role in the regulation of the sleep–wake cycle. Changes in the functioning of these systems may be involved in the pathogenesis of early stages of PD and may be age-dependent. Here, we have analyzed changes in the expression of genes associated with the regulation of the sleep–wake cycle (*Hnmt*, *Hrh1*, *Hrh3*, *Per1*, *Per2*, and *Chrm3*) in the substantia nigra (SN) and striatum of normal male mice of different ages, as well as in young and adult male mice with an MPTP-induced model of the early symptomatic stage (ESS) of PD. Age-dependent expression analysis in normal mouse brain tissue revealed changes in *Hrh3*, *Per1*, *Per2*, and *Chrm3* genes in adult mice relative to young mice. When gene expression was examined in mice with the MPTP-induced model of the ESS of PD, changes in the expression of all studied genes were found only in the SN of adult mice with the ESS model of PD. These data suggest that age is a significant factor influencing changes in the expression of genes associated with sleep–wake cycle regulation in the development of PD.

## 1. Introduction

Parkinson’s disease (PD) is a widespread neurodegenerative disease that progresses with age. A pathological feature of PD is the loss of dopaminergic (DAergic) neurons in the substantia nigra (SN) pars compacta [1]. The loss of DAergic neurons causes the clinical motor symptoms of PD, such as tremor, rigidity, bradykinesia, and postural instability [2]. It should be noted that the manifestation of the disease occurs only after the loss of about 70% of DAergic neurons, with neuronal degeneration itself developing many years prior to overt motor symptoms [3]. During this period, a range of nonmotor symptoms can manifest, among which sleep disturbances occupy a significant place [4].

Sleep disturbances (such as insomnia, excessive daytime sleepiness, sleep fragmentation, rapid eye movement (REM) sleep behavior disorder (RBD), and restless legs syndrome) may affect 40–90% of patients with PD [5]. The central histaminergic (HAergic) system plays an important role in the regulation of the sleep–wake cycle [6]. It has been shown that disruption of histamine-mediated neurotransmission has a somnogenic effect [7]. At the same time, during wakefulness, the activation of HAergic neurons of the tuberomammillary nucleus (TMN) and an increase in extracellular histamine levels were observed [8]. Histamine, in turn, can either stimulate wakefulness through interaction with HAergic H1 receptors (HRH1) [9,10] or, through binding to HAergic H3 receptors (HRH3), lead to a decrease in its own levels and the inhibition of neuronal excitation [7,11].

Currently, there is evidence that HAergic neurons, like DAergic neurons, have a number of properties that make them vulnerable to aging and the development of neurodegenerative processes. These neurons are predominantly unmyelinated and have long branched axons [12,13]. Such neuronal architecture requires significant energy expenditure, making HAergic neurons particularly vulnerable to oxidative stress.

The HAergic fibers emanating from the TMN project in particular to the SN and striatum [14], areas that are most susceptible to neuronal degeneration in PD. At present, there is some evidence of changes in the HAergic system in PD: increased levels of histamine have been found in blood and postmortem brain samples in PD patients [15,16,17,18], and changes in the expression of genes associated with HAergic neurotransmission have been observed [16,17].

It is known that the HAergic system is interconnected with circadian rhythms (physiologic and behavioral cycles with a recurring periodicity of approximately 24 h) at the molecular level [19,20]. Circadian rhythms are controlled by the circadian clock genes, that create complex interlocked transcriptional–translational feedback loops and regulate the diurnal rhythms of expression of many other genes [21]. Changes in the expression of some circadian genes have been previously identified in PD patients [22,23,24], but the role of these genes in the early stages of the disease is still unclear.

In the present work, we analyzed changes in the expression of genes associated with the regulation of the sleep–wake cycle in the SN and striatum of mice with an MPTP-induced model of the early symptomatic stage (ESS) of PD. The genes investigated in this work were selected based on our previous whole-transcriptome analysis in the brain tissue of mice with the MPTP model of the ESS of PD [25], as well as analysis of literature data. These are the *Hnmt*, *Hrh1*, and *Hrh3* genes associated with HAergic signal transduction. It was previously shown in mice that knockout of the *Hnmt* gene leads to prolonged bouts of awakening during the light (inactive) period [26]; knockout of the *Hrh1* gene leads to delayed onset of nocturnal activity [27]; knockout of the *Hrh3* gene leads to flattening of the amplitude of free-running activity rhythm [28]. We also investigated the circadian genes *Per1* and *Per2* involved in the homeostatic regulation of sleep [29], and the *Chrm3* gene, which has previously been shown to be involved in maintaining REM sleep continuity [30]. Since it is known that the functioning of the HAergic and circadian systems undergoes changes with age [31,32], it is necessary to investigate these systems in mice of different ages. Therefore, in this study, modeling of ESS PD was performed using 5- to 6-week-old and 6-month-old mice.

## 2. Results

In the present study, we analyzed changes in the expression of genes associated with regulation of the sleep–wake cycle (*Hnmt*, *Hrh1*, *Hrh3*, *Per1*, *Per2*, and *Chrm3*) in the SN and striatum of mice of different ages in normal mice, and in young and adult mice with the MPTP model of the ESS of PD.

In the first step, we analyzed age-dependent changes in gene expression in normal mice (Table 1, Figure 1). For this analysis, groups of control young and control adult mice were taken and the values for the adult mice were calculated relative to the values in the group of young mice. It was shown that a decrease in the expression of *Per1* and *Per2* genes in the SN and striatum, as well as *Hrh3* and *Chrm3* genes in the striatum was observed in adult mice relative to young mice. At the same time, no significant expression changes were obtained for the *Hnmt* and *Hrh1* genes.

In the second step, we analyzed changes in gene expression in brain samples of young and adult mice with an MPTP-induced model of the ESS of PD (Table 2, Figure 2). The values for adult mice with the MPTP model of the ESS of PD were calculated relative to the values in the group of control adult mice; the values for young mice with the MPTP model of the ESS of PD were calculated relative to the values in the group of control young mice. The data presented in Table 2 show that, in young mice with the MPTP model of the ESS of PD, a significant change in expression was observed for only one gene (decreased expression of the *Per1* gene in the striatum). In adult mice with the MPTP model of the ESS of PD, there was a significant increase in the expression of all studied genes in the SN, as well as the *Per1* gene in the striatum.

## 3. Discussion

Sleep disturbances are the most common group of nonmotor symptoms that occur in PD. These include changes in sleep architecture, as well as specific disorders such as restless legs syndrome, RBD, and sleep-disordered breathing [33]. Sleep problems in PD are commonly associated with movement disorders, nocturia, neuropsychiatric symptoms, and medications, but they can also be an inherent feature of the disease itself [34].

Regulation of the sleep–wake cycle is processed by complex interactions between multiple brain regions and neurotransmitters. One such neurotransmitter is histamine, which plays a key role in this process through binding to the HAergic receptors HRH1 or HRH3 [35]. Brain histamine is concentrated in HAergic neurons whose bodies are located in the TMN. The HAergic fibers emanating from the TMN project to virtually all parts of the brain, including the striatum and the SN [36]. At present, there is evidence that the functioning of the central HAergic system may be altered depending on age and the presence of neurodegenerative diseases [31].

It is known that the HAergic system is interconnected with circadian rhythms at the molecular level. On the one hand, it has been shown that knockout of the circadian gene *Bmal1* in mice resulted in increased expression of the *Hdc* gene encoding the enzyme histidine decarboxylase responsible for histamine synthesis. Thereby, these mice had increased histamine levels and also developed various sleep disorders [20]. At the same time, it has been shown that knockout of the *Hdc* gene leads to changes in the daily mRNA levels of the circadian genes *Per1* and *Per2* in the cortex and striatum [19]; i.e., the expression levels of these genes depend on histamine levels.

In the present study, we analyzed changes in the expression of genes associated with the regulation of the sleep–wake cycle in mouse brain samples depending on age, as well as in young and adult mice with the MPTP model of the ESS of PD. Figure 3 shows the network of functional interactions between the genes that were selected for this study (*Hnmt*, *Hrh1*, *Hrh3*, *Per1*, *Per2*, and *Chrm3*). This network shows that changes in the expression of the *Hnmt*, *Hrh1*, *Hrh3*, and *Chrm3* genes can lead to changes in histamine and dopamine levels. In turn, changes in the levels of these neurotransmitters can influence the circadian genes *Per1* and *Per2*, circadian rhythms, and PD.

### 3.1. Age-Dependent Changes in the Expression of Genes Associated with the Regulation of the Sleep–Wake Cycle in Normal Mice

Age-related changes in sleep duration are observed across the lifespan, even in the absence of pathology. Normally, sleep becomes increasingly fragmented after adolescence, and older adults develop a variety of sleep disorders [37]. These changes may be related to the fact that age affects the functioning of the HAergic system and circadian rhythms involved in the regulation of the sleep–wake cycle [31,32]. Aging people showed increased levels of histamine metabolites in the cerebrospinal fluid [38] and decreased histamine receptor HRH1 binding in the brain, predominantly in the prefrontal, temporal, cingulate, and parahippocampal regions [39,40]. Studies in rodents have shown age-related increases in the activity of HAergic neurons [41]; changes in histamine levels in the hypothalamus, midbrain, and cortex [42,43]; and changes in the concentration of proteins associated with the HAergic system [42,44] and mRNA levels of genes that encode them in various brain tissues [45].

As shown in Table 1 and Figure 1, we obtained a significant decrease in the expression of the *Hrh3* gene encoding the HAergic receptor H3 in the striatum of adult mice. Activation of the HRH3 receptor leads to a decrease in histamine release [46]. Accordingly, decreased expression of *Hrh3* may lead to increased histamine levels in adult mice and consequent stimulation of wakefulness and altered circadian rhythms (Figure 4). Our data are consistent with the results obtained in the work of Terao et al., where a significant decrease in *Hrh3* expression was observed in the medulla of mice with increasing age. In this work, a tendency for a decreased level of *Hrh3* expression in the basal forebrain was also observed, but the results obtained for this region were not statistically significant [45]. We found reduced expression of *Hrh3* in the striatum of adult mice compared to young mice for the first time.

It is also known that the functioning of the circadian system changes with age [47,48,49,50]. In particular, there are changes in the daily rhythms of physiological parameters [51,52,53], as well as changes in the rhythmicity and expression level of circadian genes [54,55,56,57,58,59]. In the present study, we found decreased expression of the *Per1* and *Per2* genes in the SN and striatum of adult mice relative to young mice (Table 1, Figure 1). These genes encode Period1 (PER1) and Period2 (PER2) proteins, which are the main molecular components of circadian rhythms and regulate the circadian rhythms of motor activity, metabolism, and behavior [60]. The observed decrease in *Per1* and *Per2* gene expression could be related to a temporal phase shift in circadian rhythms at the molecular level in adult mice. This effect could be caused by increased HAergic activity due to impaired inhibition of histamine release, as in our work we observed a threefold decrease in *Hrh3* expression level in adult mice (Figure 4). It is possible that the reduction in *Per1* and *Per2* gene expression in adult mice that we have revealed will persist with further increase in the age of mice. Thus, a number of studies have shown a decrease in the levels and amplitude, as well as a phase shift in the diurnal rhythm of Per1 and Per2 levels, in old rodents compared to younger rodents, both at the transcript and protein levels [55,56,61,62,63,64,65,66,67,68]. Similar age-dependent changes have been observed in humans, but there are fewer such studies [57,69,70]. Taking into account that the *Per1* and *Per2* genes are participants of the system of regulation of circadian rhythms, a significant decrease in their expression may lead to changes in the expression of target genes regulated by them [71].

Another gene for which we observed a significant decrease in expression in the striatum of adult mice was *Chrm3*, encoding the muscarinic acetylcholine receptor M3 (Table 1, Figure 1) [72]. The expression of this gene is known to be regulated by circadian rhythms [73]. In the present study, as described previously, we observed decreased expression levels of the circadian genes *Per1* and *Per2* in adult mice compared to young mice, which may have mediated the effect on *Chrm3* gene expression. Our results are consistent with previously published data. Decreased Chrm3 receptor concentration was observed in the cerebellum of rats aged 90 days relative to rats aged 16 and 21 days [74]. In addition, Sanfilippo et al. found a decrease in *CHRM3* gene expression with increasing age in healthy people without dementia [75]. The published data suggest that activation of the CHRM3 receptor may regulate the level of HAergic HRH1 receptors in two ways. On the one hand, activation of the CHRM3 receptor promotes a decrease in HRH1 receptor levels through degradation of the protein itself; on the other hand, it can lead to up-regulation of the HRH1 receptor by inducing its mRNA synthesis [76,77]. Since no change in *Hrh1* mRNA levels was detected in adult mice in our work, it can be hypothesized that decreased expression of *Chrm3* will contribute to the conservation of already existing Hrh1 receptors. In addition, *Chrm3* has previously been shown to be involved in sleep regulation. Knockout of this gene resulted in a decreased duration of nonrapid eye movement (NREM) sleep and fragmented REM sleep [30]. It is possible that decreased expression of *Chrm3* in adult mice will also lead to changes in their sleep patterns (Figure 4).

### 3.2. Changes in the Expression of Genes Associated with the Regulation of the Sleep–Wake Cycle in Young and Adult Mice with the MPTP-Induced Model of the ESS of PD

Symptoms associated with sleep disturbance are often observed in PD patients even before diagnosis. On this basis, it can be assumed that changes in systems involved in sleep regulation may be involved in the pathogenesis of this disease at the earliest stages. We analyzed changes in the expression of genes associated with the regulation of the sleep–wake cycle in the SN and striatum of mice with the MPTP-induced model of the ESS of PD. To assess the contribution of age to gene expression changes in the PD model, brain tissue samples from young and adult mice were used for analysis.

As can be seen from Table 2 and Figure 2, for all the genes under study (*Hnmt*, *Hrh1*, *Hrh3*, *Per1*, *Per2*, and *Chrm3*), a significant increase in the expression levels by 1.5 and more times in the SN of adult mice with the MPTP-induced model of the ESS of PD relative to adult healthy mice was detected. At the same time, young mice with the MPTP-induced model of the ESS of PD had no changes in the expression of the studied genes in the SN relative to young healthy mice. This result may be a consequence of the higher compensatory capacity of young mice. In addition, susceptibility to MPTP toxicity in animals increases with age [78,79,80,81,82,83,84]. Such changes may be related to the fact that with age there is an accumulation of reactive oxygen species and cell damage caused by oxidative stress [85], as well as activation of microglia [86].

Interestingly, in the striatum, changes in the expression of the genes studied, with the exception of *Per1*, were absent in adult mice with the MPTP-induced model of the ESS of PD. Figure 3 shows the interaction network of the genes studied in this work, demonstrating that all of them are characterized by links with histamine. The literature says that the SN has a strong innervation by HAergic neurons, whereas the striatum has a low-to-moderate level of innervation [36]. It can be assumed that histamine accumulates in the SN unlike the striatum in adult mice, affecting the expression of the genes under MPTP exposure.

In our work, we observed a simultaneous increase in the expression of *Hnmt* and *Hrh3* in the SN of adult mice with the ESS model of PD. There is evidence that PD patients are characterized by increased histamine levels [15,18], and elevated histamine levels may contribute to the degeneration of DAergic neurons and trigger inflammatory signaling processes [15,87]. Hnmt is a key enzyme in histamine metabolism [88], and activation of the Hrh3 receptor leads to decreased histamine release [46]. Thus, increased expression of these genes may indicate the development of compensatory mechanisms aimed at reducing histamine levels (Figure 5).

In turn, the increase in *Hrh1* gene expression observed in our work may have deleterious effects on DAergic neurons. Such an alteration may further increase histamine levels, because Hrh1 is an activating receptor that promotes histamine release [89]. Histamine, through activation of the HAergic receptor Hrh1, causes microglia activation and ultimately the death of DAergic neurons [90]. Accordingly, increased expression of *Hrh1* in the SN may lead to microglia activation and, consequently, to the development of neurodegeneration (Figure 5).

Thus, increased expression of *Hnmt* and *Hrh3* genes, which may lead to decreased histamine levels simultaneously with increased expression of *Hrh1*, which, on the contrary, may lead to increased histamine levels, indicates the complex nature of changes in the functioning of the HAergic system during the development of ESS of PD.

It is noteworthy that young mice with the model of the ESS of PD showed a decrease in circadian *Per1* gene expression in the striatum, whereas, in adult mice with the model of the ESS of PD, expression of this gene increased both in the striatum and in the SN (Table 2, Figure 2). This result may be related to the fact that MPTP affects the phase shift of diurnal *Per1* expression, and the direction of this phase shift is age-dependent. In addition, an increase in circadian *Per2* gene expression in the SN was observed in adult mice with the model of the ESS of PD. Published data suggest that Per2 is involved in the regulation of autophagy and also modulates cell death in response to oxidative stress [57,91]. Therefore, it cannot be definitively stated whether increased expression of this gene in the SN of adult mice with the model of the ESS of PD would contribute to the death of DAergic neurons in response to MPTP exposure or have a compensatory effect by triggering autophagy. However, our findings indicate an age-dependent involvement of the *Per2* gene in the development of the early stages of PD and emphasize the need for a more detailed study of its role in the pathogenesis of the disease. In general, it can be assumed that the detected changes in the expression of the *Per1* and *Per2* genes may lead to the development of symptoms associated with circadian rhythm disturbances at the early stages of PD pathogenesis (Figure 5).

In our work, the *Chrm3* gene in the SN of adult mice with the ESS model of PD had the greatest change in expression among all genes. The projections of cholinergic neurons from the pedunculopontine nucleus (PPN) are directed to the SN [92]. It is known that PPN cholinergic neurons undergo degeneration in PD patients [93]. Cholinergic denervation is associated with one of the earliest symptoms of the disease, namely RBD [94]. It has been previously shown that knockout of the *Chrm3* gene leads to fragmentation of REM sleep [30]. On this basis, we hypothesize that the increased expression of *Chrm3* in our work may be a compensatory response to the degeneration of cholinergic neurons and will have a beneficial effect on the nonmotor symptoms of PD patients, contributing to the maintenance of REM sleep continuity in them (Figure 5).

On the other hand, Zhang et al. showed that activation of the Chrm3 receptor suppresses dopamine release in the striatum [95]. Thus, it can be supposed that the increase in *Chrm3* expression observed in mice with the model of the ESS of PD may lead to suppression of DAergic neuronal activity. In earlier studies, an increase in M3 receptor binding was found in postmortem samples of the striatum and dorsolateral prefrontal cortex of PD patients [96].

Thus, our observed increase in *Chrm3* expression may be related to the involvement of the cholinergic system in neurodegeneration in PD. At the same time, since the increase in *Chrm3* expression was observed only in adult mice with the model of the ESS of PD, it can be assumed that the involvement of the cholinergic system is age-dependent. This is also indicated by the results of the work of Karachi et al., where, in a study of young and old macaques with the MPTP-induced model of PD, death of DAergic neurons was observed in both age groups of animals, but death of cholinergic neurons of the PPN and postural instability were observed only in old macaques [97].

## 4. Materials and Methods

### 4.1. Modeling PD

In this work, the MPTP model of the ESS of PD was used. This model was established according to the protocol described earlier [98]. Male C57BL/6 mice were divided into four study groups (n = 10 in each group): a control group of young mice (age 5–6 weeks), a control group of adult mice (age 6 months), an experimental group of young mice (age 5–6 weeks), and an experimental group of adult mice (age 6 months). The animals were kept in a vivarium with free access to food and water and under natural light. All efforts were made to minimize the potential suffering and discomfort of animals and care was taken according to the 3Rs (replacement, reduction, and refinement) [99]. The study was approved by the Ethics Committee of the Institute of Molecular Genetics of the National Research Centre “Kurchatov Institute” (protocol code 3/21, 17 February 2021).

MPTP was administered to mice from the experimental groups (Sigma-Aldrich, St. Louis, MO, USA) four times at an individual dose of 12 mg/kg of body weight with an interval of 2 h. Saline (0.9% NaCl) was administered to control animals according to a similar scheme [98]. Mice were sacrificed 2 weeks after the last injection. The brain was removed from the skull and cut along the midsagittal plane. The striatum and SN were then excised from the brain according to the mouse brain atlas [100]. Tissue samples were weighed, fresh frozen in liquid nitrogen, and stored at −70 °C until they were used for RNA isolation.

### 4.2. RNA Isolation and Expression Analysis of Individual Candidate Genes

RNA isolation from tissue samples of each animal and analysis of mRNA levels by reverse transcription and real-time PCR (TaqMan technology, QuantStudio 3 cycler, Applied Biosystems, Waltham, MA, USA) were performed according to protocols described previously [101].

Primer and probe sequence design was produced using the Beacon designer 7.0 software (Premier Biosoft International, Palo Alto, CA, USA) and nucleotide sequences of the candidate genes *Hnmt*, *Hrh1*, *Hrh3*, *Per1*, *Per2*, and *Chrm3*, and housekeeping genes *Sars1* and *Psmd6* from the NCBI database [102]. The sequences of gene-specific primers and probes are presented in Table 3.

### 4.3. Statistical Processing of Data

The statistical and bioinformatics analysis protocol is described in detail in work previously performed in our laboratory [101]. Relative gene-expression levels in patient samples were calculated using the Comparative Ct Method ΔΔCt [103]. The nonparametric Mann–Whitney *U* test was used to evaluate relative gene expression levels.

## 5. Limitations of the Study

As far as we know, the effect of age on changes in the expression of genes associated with the regulation of the sleep–wake cycle in mice with the MPTP-induced model of the ESS of PD was studied by us for the first time. It is worth noting that PD modeling using MPTP is preferentially performed in young mice around 8 weeks of age [98,104,105,106] despite the fact that PD is an age-dependent disease [2]. Our study has convincingly demonstrated that the expression profiles of the studied genes will differ even in young and adult mice with the PD model. However, for a more detailed study, it is necessary to increase the number of groups of mice of different ages and, first of all, to study old animals. Taking age into account will allow a better interpretation of the transcriptome changes that are caused by the development of PD.

Another limitation of our study is that it was conducted solely at the transcriptome level. Our findings characterize changes at the mRNA level and, in interpreting the data, we relied in part on the known functions of the proteins encoded by the studied genes. Future work should include studies at the protein level, as well as measurements of the levels of neurotransmitters mentioned in this study (histamine and dopamine).

Finally, in the present study, we investigated whole tissue samples of the striatum and SN. These regions consist of different populations of neurons, as well as glial cells, and the contribution to changes in gene expression may differ between these cell types. Assessing this contribution could undoubtedly provide a more complete picture of the processes occurring under the influence of age and the development of a Parkinson’s-like phenotype, and is a promising direction for future research.

## 6. Conclusions

In the present study, we analyzed changes in the expression of genes associated with the regulation of the sleep–wake cycle in the SN and striatum of mice of different ages in normal mice, as well as in young and adult mice with the MPTP model of the ESS of PD.

Age-dependent expression analysis in mouse brain tissues revealed changes in the expression of the *Hrh3*, *Per1*, *Per2*, and *Chrm3* genes in adult mice relative to young mice. These results confirm that changes in the functioning of the HAergic and cholinergic systems and circadian rhythms occur with age.

When gene expression was examined in mice with the MPTP-induced model of the ESS of PD, changes in the expression of the *Hrh1*, *Hrh3*, and *Hnmt* genes associated with HAergic signal transduction, as well as the *Chrm3* gene encoding the cholinergic muscarinic receptor, were found in the SN of adult mice with the ESS model of PD. The findings once again point to the complexity of neurodegenerative processes in this disease, as well as the involvement of different neurotransmitter systems in the development of PD at the early stages of pathogenesis. In addition, altered expression of the circadian genes *Per1* and *Per2* in mice with the model of the ESS of PD suggests that these genes may be involved in the development of symptoms associated with circadian rhythm disruption in patients in the early stages of the disease.

Overall, our findings suggest that the expression of genes associated with sleep–wake cycle regulation (*Hnmt*, *Hrh1*, *Hrh3*, *Per1*, *Per2*, and *Chrm3*) in the SN is altered in adult mice with the model of the ESS of PD and unchanged in young mice with the model of the ESS of PD. On this basis, we conclude that age is a significant factor influencing changes in the expression of genes associated with sleep–wake cycle regulation in the development of a Parkinson-like phenotype. Also, the findings suggest that changes in circadian, HAergic, and cholinergic systems have complex interactions in the ESS of PD. However, this hypothesis should be addressed in future studies.

## Figures and Tables

**Figure 1 ijms-25-07721-f001:**
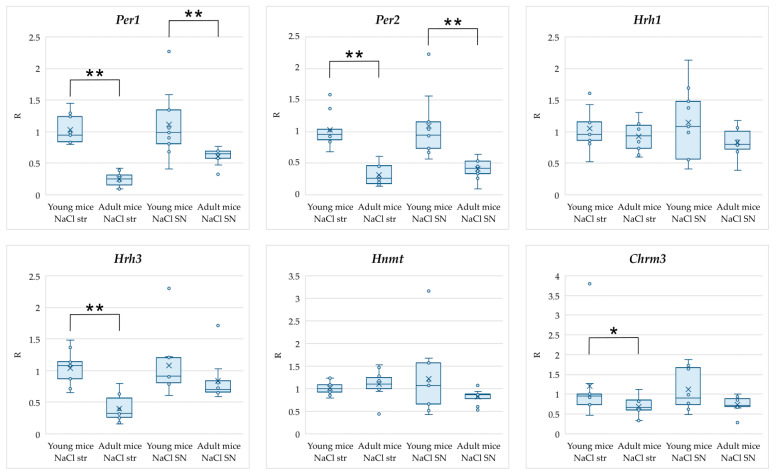
Results of analysis of changes in expression of genes associated with regulation of the sleep–wake cycle in the striatum and SN of adult mice (6 months old) relative to young mice (5–6 weeks old) presented as bar plots showing the individual data points. Abbreviations: str—striatum; SN—substantia nigra; NaCl—control mice. * *p* < 0.05, ** *p* < 0.01.

**Figure 2 ijms-25-07721-f002:**
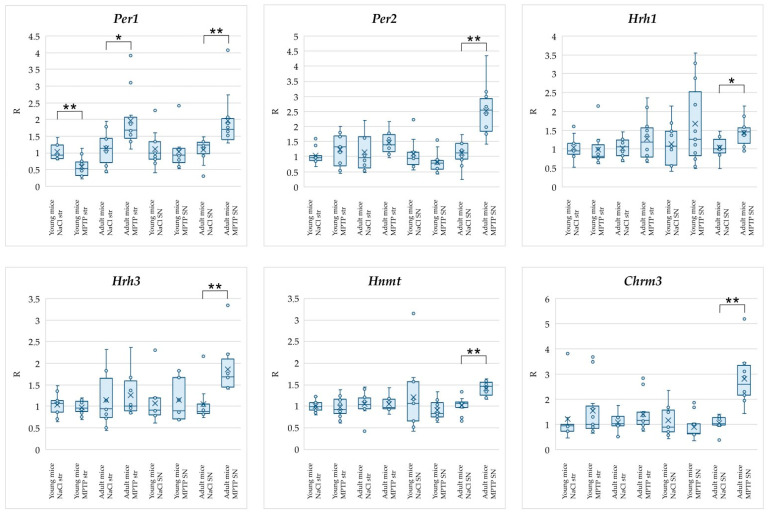
Results of analysis of changes in expression of genes associated with regulation of the sleep–wake cycle in the striatum and SN of mice with MPTP-induced model of the ESS of PD presented as bar plots showing the individual data points. Abbreviations: str—striatum; SN—substantia nigra; NaCl—control mice; MPTP—mice with MPTP-induced model of the ESS of PD. * *p* < 0.05, ** *p* < 0.01.

**Figure 3 ijms-25-07721-f003:**
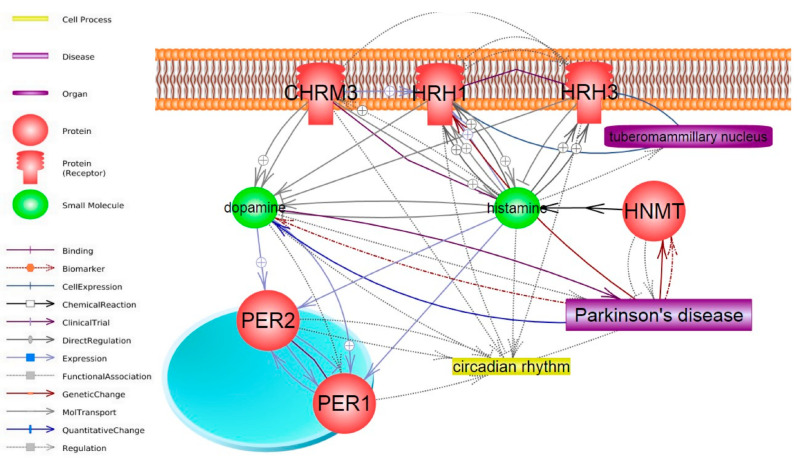
Interaction network of the studied genes. The network was constructed using Pathway Studio v. 12.4.0.5.

**Figure 4 ijms-25-07721-f004:**
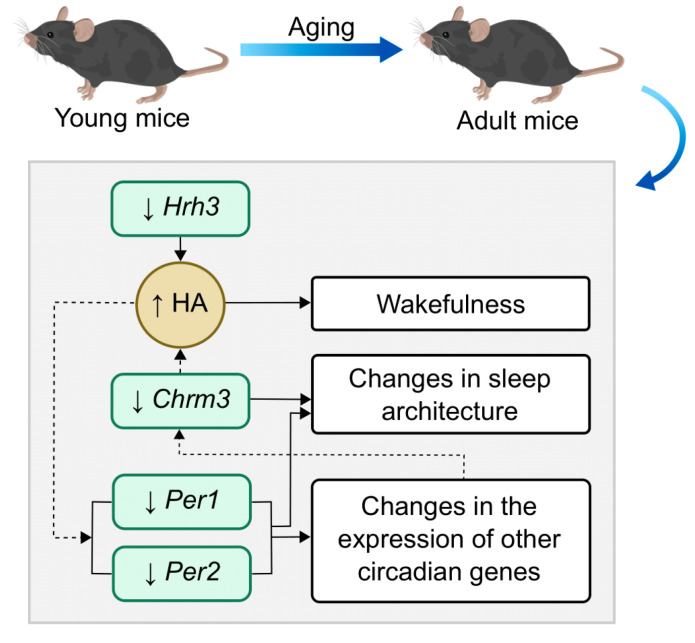
Schematic showing the possible impact of changes in the expression of the genes associated with regulation of the sleep–wake cycle during aging. Solid lines indicate interactions confirmed in the literature. Dotted lines indicate possible interactions. Up arrows (↑) indicate an increase, down arrows (↓) indicate a decrease. Abbreviation: HA—histamine.

**Figure 5 ijms-25-07721-f005:**
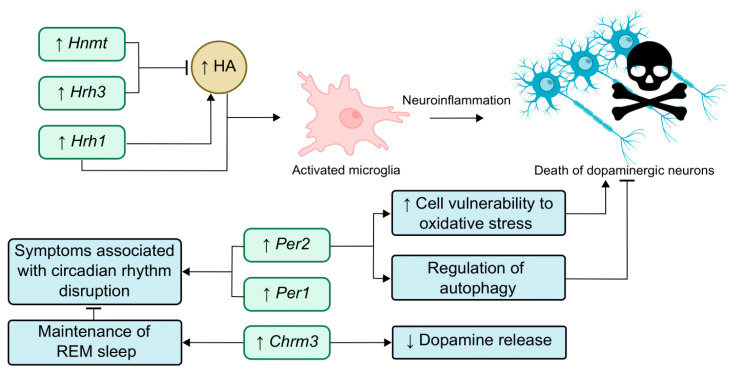
Schematic showing the possible impact of changes in the expression of the genes associated with regulation of the sleep–wake cycle on the ESS of PD in adult mice. Sharp arrows (→) indicate stimulation, blunt arrows (┴) indicate inhibition, up arrows (↑) indicate an increase, down arrows (↓) indicate a decrease Abbreviations: HA—histamine; REM—rapid eye movement.

**Table 1 ijms-25-07721-t001:** Results of analysis of changes in expression of genes associated with regulation of the sleep–wake cycle in the striatum and SN of adult mice (6 months old) relative to young mice (5–6 weeks old).

Gene	Striatum	SN
*Per1*	**0.24 ^1^** **0.15–0.31 ^2^**	**0.65** **0.58–0.69**
*Per2*	**0.26** **0.17–0.46**	**0.41** **0.34–0.53**
*Hnmt*	1.10 0.99–1.25	0.86 0.78–0.88
*Hrh1*	0.93 0.74–1.10	0.80 0.72–1.01
*Hrh3*	**0.32** **0.26–0.56**	0.70 0.66–0.84
*Chrm3*	**0.66** **0.61–0.85**	0.72 0.68–0.89

^1^ Median, ^2^ 25–75 percentiles. Data in bold and gray background are statistically significant (*p* < 0.05). The expression level in the group of young mice is taken as 1.

**Table 2 ijms-25-07721-t002:** Results of analysis of changes in expression of genes associated with regulation of the sleep–wake cycle in the striatum and SN of mice with MPTP model of the ESS of PD.

	Young Mice		Adult Mice	
Gene	Striatum	SN	Striatum	SN
*Per1*	**0.52 ^1^** **0.32–0.73 ^2^**	0.92 0.71–1.13	**1.67** **1.44–2.06**	**1.70** **1.39–2.02**
*Per2*	1.32 0.68–1.68	0.79 0.57–0.88	1.39 1.18–1.73	**2.54** **1.83–2.92**
*Hnmt*	0.93 0.83–1.17	0.84 0.75–1.09	0.97 0.94–1.16	**1.47** **1.26–1.57**
*Hrh1*	0.81 0.76–1.12	1.26 0.82–2.53	1.18 0.78–1.57	**1.46** **1.14–1.56**
*Hrh3*	0.96 0.88–1.12	0.91 0.70–1.67	1.01 0.90–1.59	**1.68** **1.45–2.09**
*Chrm3*	1.13 0.91–1.75	1.25 1.04–1.62	1.15 0.99–1.47	**2.60** **2.16–3.36**

^1^ Median, ^2^ 25–75 percentiles. Data in bold and gray background are statistically significant (*p* < 0.05). The expression level in the control groups is taken as 1.

**Table 3 ijms-25-07721-t003:** Sequences of gene-specific primers and probes.

Gene (Protein)	Nucleotide Sequence
*Sars1* (Seryl-aminoacyl-tRNA Synthetase) NM_011319.3 *	Probe: 5′-VIC-CGTTCTACTTTGTTGTCTGCGTCCTCATCA-BHQ2-3′ Forward primer: 5′-GCGAGATTGGGAACCTTCTG-3′ Reverse primer: 5′-ATGGGAATACTTCTTCCTGACTGTA-3′
*Psmd6* (Proteasome 26S subunit, non-ATPase, 6) NM_025550.2 *	Probe: 5′-VIC-AATCGTGGAGACCAACAGACCTGATAGCAA-BHQ2-3′ Forward primer: 5′-GGTGTGGGTGTGGACTTCATT-3′ Reverse primer: 5′-CTCCTTTCTTGATGGTTTCTTGATACTG-3′
*Hnmt* (Histamine N-methyltransferase) NM_080462.2 *	Probe: 5′-VIC-CAACTTCACCTGCACCTCCGCCTACACTCA-BHQ2-3′ Forward primer: 5′-AAGGATTGGAGAAGCAAAAGCAG-3′ Reverse primer: 5′-TGTTCAGCACTTGGCTCAAC-3′
*Hrh1* (Histamine H1 Receptor) NM_001252642.2 *	Probe: 5′-VIC-TGATGGCTCCCTCCCTCGGTCTCTGGC-BHQ2-3′ Forward primer: 5′-GCTACTGTGGGCTGGTGATTC-3′ Reverse primer: 5′-AGGTGTTGGGAAGGCTCATTG-3′
*Hrh3* (Histamine H3 Receptor) NM_133849.3 *	Probe: 5′-VIC-TCCGACTTCCTCGTGGGTGCCTTC-BHQ2-3′ Forward primer: 5′-CTGGTCATGCTCGCCTTCG-3′ Reverse primer: 5′-CGGTCAGCACATAGGGTACATAC-3′
*Per1* (Period Circadian Clock 1) NM_011065.5 *	Probe: 5′-VIC-TCGTGGACTTGACACCTCTTCTGTGGC-BHQ2-3′ Forward primer: 5′-CCTTCCTCAACCGCTTCAG-3′ Reverse primer: 5′-CGGGAACGCTTTGCTTTAGA-3′
*Per2* (Period Circadian Clock 2) NM_011066.3 *	Probe: 5′-VIC-TGTTTCCCAACACTGACACGGCAGAAA-BHQ2-3′ Forward primer: 5′-GCGGCTTAGATTCTTTCACTCA-3′ Reverse primer: 5′-ATGCGGAAGGGCTGGTAG-3′
*Chrm3* (Cholinergic Receptor Muscarinic 3) NM_033269.4 *	Probe: 5′-VIC-GAGTGAACCATATCCTTTCCCATCA-BHQ2-3′ Forward primer: 5′-GCCCTTATTGTACCTTTGCTGAAG-3′ Reverse primer: 5′-CTCCTCTTGAAGTGCTGCGTTCTGACC-3′

* Accession numbers in the GenBank database (NCBI GenBank Release 256.0). VIC—fluorescent dye; BHQ2—fluorescence quencher.

## Data Availability

The datasets used during the current study are available from the corresponding author on reasonable request.

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
