# Peer review of "Effects of Age and MPTP-Induced Parkinson’s Disease on the Expression of Genes Associated with the Regulation of the Sleep–Wake Cycle in Mice"

_ijms, 2024, doi:10.3390/ijms25147721_

Round 1
Reviewer 1 Report
Comments and Suggestions for Authors
The study of Semenova et al. was interesting and well-written. It certainly has the big limit of having made only a relative and not absolute transcriptomic evaluation. The work deserves to be improved in order to be published. in particular I have raised some critical issues:
- Authors make too much reference to their previous studies for methods forcing the reader to search in other papers the information. The authors should explain better especially the experimental design and the methods in this manuscript by treating it as a stand-alone allowing the reader to find all the useful information.
-I suggest to move the box plot from supplementary to the main figure of results.
-Also, can authors explain better tha statistical analysis performed for each comparison? Why they used Mann-Whitney test? Why authors represent the median of gene expression and not the average considering that young mice or control groups were taken as 1?
-authors should better indicate the limits of the work. Further different analyses are needed to really assert an association with neurotransmitters.
Comments on the Quality of English LanguageMinor editing of English language required.
Reviewer 2 Report
Comments and Suggestions for Authors
Parkinson's disease (PD) is a progressive neurodegenerative disorder characterized by motor symptoms such as tremors, rigidity, bradykinesia, and postural instability. However, non-motor symptoms, including sleep disorders, are also prevalent and can significantly impact the quality of life for individuals with PD. Sleep disturbances in PD are multifactorial, stemming from both the disease itself and the medications used to manage its symptoms. This manuscript aims to examine the expression of genes associated with the regulation of the sleep-wake cycle of mice substantia nigra (SN) and striatum region. The comparation include young vs. adult and adult vs. MPTP, these data increased our knowledge about the sleep-wake cycle in the SN and striatum of mice with early symptomatic stage (ESS) of PD. However, there are several major and minor weaknesses in the rationale and research methods of this work. Below please find the review comments.
(1) Major comments
1. The biggest limitation of the study is that the evidence is not sufficient to support the conclusion, the QPCR analysis of mRNA levels is just one of the techniques that are commonly used to measure and analyze gene expression changes. It is essential to validate the QPCR results by other independent methods such as Western blotting for protein validation or in situ hybridization for spatial expression patterns to ensure consistency and reliability. Moreover, the global gene expression changes within the striatum and SN samples are somewhat too gross, it is better to measure the cell type specific changes.
2, It was unclear why the authors chose these genes (Hnmt, Hrh1, Hrh3, Per1, Per2, Chrm3) for analysis, as the sleep-wake cycle is regulated by a complex interplay of genetic, neural, and environmental factors. Numerous genes have been identified that play critical roles in various aspects of sleep regulation, including circadian rhythms, sleep homeostasis, and neurotransmitter systems, the evidence in line 73-74 is not convincible, the related literature should be added or listed to support why this gene is essential to sleep-wake cycle, and what does it implies after the gene expression is changed.
3. It appears that the author uses mice of 5- to 6-week-old and 6-month-old to study the age-dependent changes of (sleep-wake cycle) gene expression. However, I do not think 2 groups (young and adult) are sufficient to summarize this conclusion, more mice with different age are required to support the conclusion. Based on the Jackson Lab (https://www.jax.org/news-and-insights/jax-blog/2017/november/when-are-mice-considered-old), 6 months old mice are equivalent to 30 years old human, but there is a little ratio of (30 years old) human get PD or sleep-wake disorders. Please clarify these points when interpreting the data.
4, In adult mice with the MPTP models, the author merely examined the MPTP model vs. control adult. It is essential to detect the correlation of sleep-wake cycle genes with the features of MPTP PD models, whether the decreased expression of the Per1 gene in the striatum is correlated with MPTP-induced PD pathophysiology? For example, how the behavior deficient of sleep-wake cycle within MPTP PD models? It would be beneficial to clarify the correlations of sleep-wake cycle gene changes with the phonotype of sleep-wake cycle within MPTP PD models.
(2) Minor comments
1. Upon reviewing the manuscript, it is important to revise the title to make it short and brief, the current version is too long to get the main points.
2. Abbreviations should be properly defined and interpreted when they are first introduced in the manuscript. This practice ensures that readers can understand the meaning of abbreviations without confusion. It is also recommended to include a comprehensive list of abbreviations after the conclusion section. This list will serve as a quick reference for readers, enabling them to easily access the definitions throughout the manuscript.
3. Need citations to indicate the dosage and treatment frequency of MPTP in the methods sections (line 92-93).
4. The method section showed that male C57BL/6 mice is used for this study, it is important to state the sex of the samples in the title and abstract of the paper, as sex is a biological variable factor and associated with physical and physiological features, especially in the PD studies, as men are more likely to develop PD than women. Please add the reason to use male mice only for this study, otherwise, a limitation in the discussion section is required.
5. The authors should make the necessary modifications to the format of all the tables to ensure they comply with the specific journal requirements. This may involve adjusting the table layout, font style and size, column headings, and other elements to align with the journal's guidelines.
6. Limitations of the MPTP model for PD studies should be discussed objectively.
Comments on the Quality of English LanguageMinor editing of English language required.
Round 2
Reviewer 1 Report
Comments and Suggestions for Authors
Authors have addressed my concerns satisfactorily.
Reviewer 2 Report
Comments and Suggestions for Authors
The authors have significantly revised the paper with addition of new data and have addressed all the point raised in the initial review.